# Informing future research for carriage of multiresistant Gram-negative bacteria: problems with recruiting to an English stool sample community prevalence study

Donna M Lecky,[1] Deborah Nakiboneka-Ssenabulya,[1] Tom Nichols,[2] Peter Hawkey,[3] Kim Turner,[1] Keun-Taik Chung,[3] Mike Thomas,[4] Helen Lucy Thomas,[5] Li Xu McCrae,[3] Sahida Shabir,[3] Susan Manzoor,[3] Adela Alvarez-Buylla,[3] Steve Smith,[6] Cliodna McNulty[1]

DML and DN-S contributed equally.

For numbered affiliations see end of article.

**Correspondence to**
Dr Donna M Lecky;
Donna.lecky@phe.gov.uk

## ABSTRACT

**Objectives** This study aims to highlight problems with recruiting to an English stool sample community prevalence study. It was part of a larger cross-sectional research to determine the risk factors for the presence of extended-spectrum beta-lactamase and carbapenemase-producing coliforms in stool samples of the asymptomatic general English population.

**Setting** Four National Health Service primary care trusts (PCTs) of England representing a different section of the population of England: Newham PCT; Heart of Birmingham Teaching PCT; Shropshire County PCT; and Southampton City PCT.

**Participants** Sixteen general practices across the four PCTs were purposefully selected. After stratification of GP lists by age, ethnicity and antibiotic use, 58 337 randomly selected patients were sent a postal invitation. Patients who had died, moved to a different surgery, were deemed too ill by their General Practitioner or hospitalised at the time of mailing were excluded.

**Results** Stool and questionnaire returns varied by area, age, gender and ethnicity; the highest return rate of 27.3% was in Shropshire in the age group of over 60 years; the lowest, 0.6%, was in Birmingham in the age group of 18–39 years. Whereas only 3.9%(2296) returned a completed questionnaire and stool sample, 94.9% of participants gave permission for their sample and data to be used in future research.

**Conclusion** Researchers should consider the low stool specimen return rate and wide variation by ethnicity and age when planning future studies involving stool specimen collection. This is particularly pertinent if the study has no health benefit to participants. Further research is needed to explore how to improve recruitment in multicultural communities and in younger people.

## Strengths and limitations of this study

► This was a large multi-centre community based study that included adult participants of variable age groups, gender and ethnicities from four areas in England.
► By inviting a large number of patients from different ethnicities to participate, return rates are likely to be comparable in future studies.
► Recruiting patients in batches at each practice allowed us to compensate for the lower than expected return rate by increasing invited in cohorts with lower returns.
► Use of a stool collection instruction leaflet and a pre-packaged stool kit delivered to participant's homes may have aided compliance and stool returns.
► Ethics permitted only anonymous patient information be removed from practices meaning researchers could not follow up with participants who did not respond to the initial invite; this is unfortunate as follow up phone calls and interaction with the research team encourage higher recruitment rates.

of invasive infections.[1] Researching gut carriage of multiresistant bacteria in the asymptomatic population will help inform the need for control efforts as gut organisms are a source of Gram-negative infections. We do not know if prevalence research for gut carriage of antibiotic-resistant organisms using postal stool samples is feasible; therefore, understanding the challenges associated with obtaining postal stool samples is critical to the design of population-based research studies.

Recruitment of patients to research studies where they are asked to submit stool samples can be difficult, particularly when there is no obvious benefit to the participant. At community surveillance level, the Bowel Cancer Screening

## BACKGROUND

Reports from the European Antimicrobial Resistance Surveillance Network data show that multi-drug-resistant *Escherichia coli* now comprise 15%

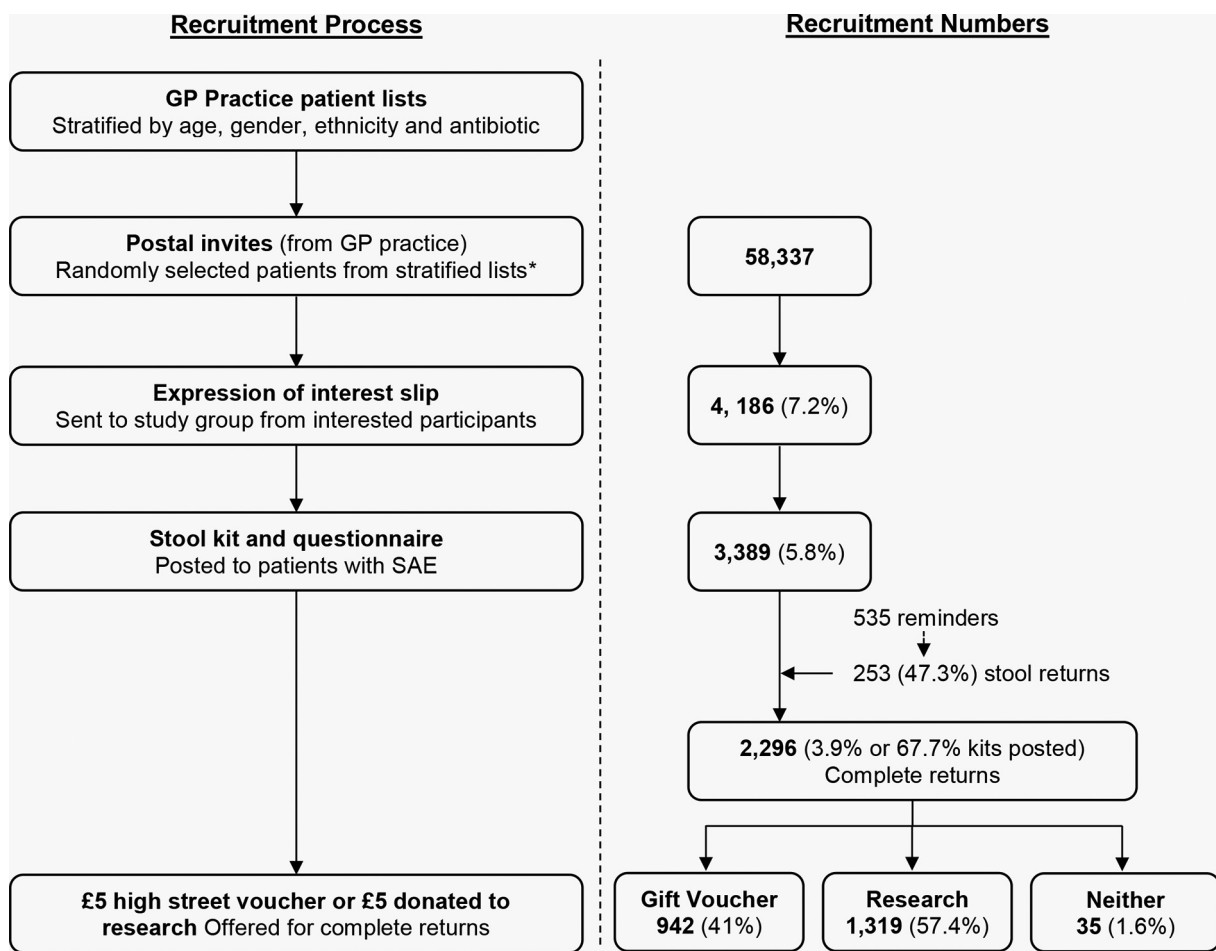

**Figure 1** Participant recruitment. SAE, self addressed envelope.

Programme (BCSP), targeting adults over 60 years of age in England, found that stool specimen returns were 54% overall but lower among the Black and Minority Ethnic (BME) groups especially within the Asian population.[2] A general lack of opportunities to engage in research and cultural or religious practices have previously been highlighted as barriers to ethnic minority participation.[2–5] Problems with community recruitment can occur at different stages of the process, for example, obtaining patient lists, stratifying the data, obtaining consent, drop out following consent, and so on.[6 7] but there is little information on population studies in asymptomatic individuals.

This paper aims to describe challenges faced when obtaining self-collected stool samples and self-administered questionnaires from healthy participants invited and recruited by post. It examines how stool return rate varied between different ethnic groups, age group, gender and the four National Health Service (NHS) primary care trusts (PCTs) selected. This research will inform future surveys using stool specimens.

## MATERIALS AND METHODS
### PCT selection
Four NHS PCTs of England were selected non-randomly to represent a different section of the population

of England: Newham PCT (London, urban, relatively high proportion of South Asian, Caribbean and African patients); Heart of Birmingham Teaching PCT (urban, very high proportion of South Asian patients); Shropshire County PCT (rural, very high proportion of White-British patients); and Southampton City PCT (semi-urban, high proportion of White-British and also a relatively high proportion of South Asian patients). Ethnicity data for each PCT were taken from Population Estimates by Ethnic Group in England,[8] while the Index of Multiple Deprivation for each practice was determined from online General Practitioner (GP) Practice Profiles.

### GP practice selection
We worked with Primary Care Research Networks (PCRN) to facilitate recruitment of practices. All practices in a PCRN were invited by letter to participate. As ethnicity was a key criterion for patient selection, practices were excluded if they had not recorded ethnicity for at least 50% of their patients. Four or five practices that were willing to participate and were from the PCRN of each PCT were non-randomly selected to broadly represent each PCT with respect to ethnicity and deprivation. Overall 16 practices were recruited to the study: three from Shropshire, four from Newham, five from Southampton and four from Birmingham.

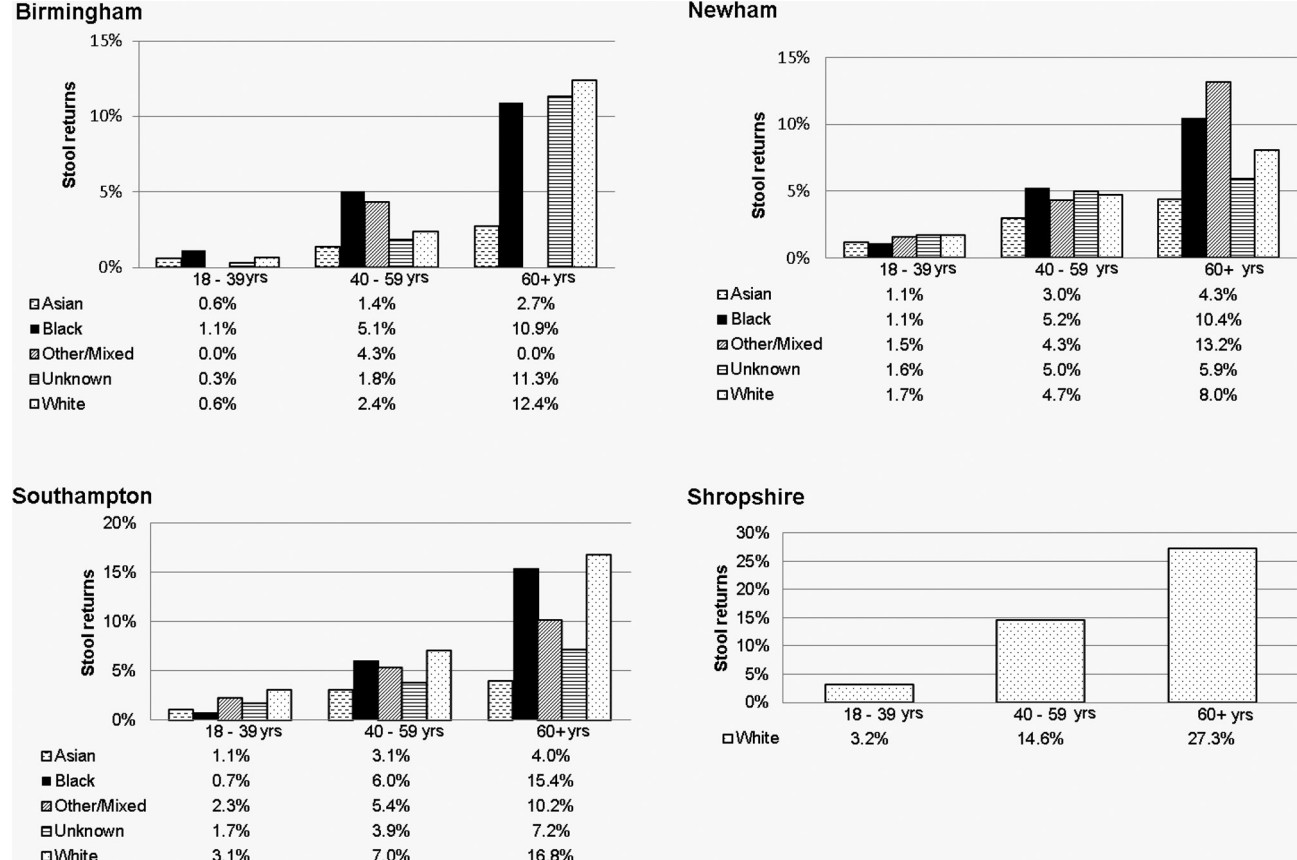

**Figure 2** Participant demographics by stool return rate.

## Patient selection

The study aimed to recruit 390 participants from each specific ethic group (black, white, Asian, mixed, unknown) across the four PCTs. Patients in selected practices aged 18 years and above were stratified by a number of factors, including GP record of ethnicity, gender, age and antibiotic use in the previous year.

## Patient screening by practice clinician

Patient lists were screened by a practice clinician to check suitability for inclusion. Excluded participants included those who died, moved to a different surgery, were deemed too ill by their GP or hospitalised at the time of mailing.

## Patient invitation process

Between November 2013 and October 2014, stratified lists were randomised and patients invited in order from these lists. A disproportionately large number of invites were sent to patients from those strata containing ethnic minority group patients (disproportionate stratified random sampling). Patients received an invitation letter containing a sentence in English, and in four of the most commonly used non-English languages spoken in that GP practice, inviting them to request a translation of the study information in their preferred language (figure 1). Letters explained that

- ▶ the main study aimed to find out what things made some people more likely to carry different bacteria in the gut
- ▶ if they agreed to participate, they would be asked to return a stool specimen and a short questionnaire about things that may affect bacteria in the gut such as antibiotic use, hospital visits, diet and travel
- ▶ information would be kept confidential
- ▶ they could opt out of the study at any time
- ▶ participants would be given the option to receive either a £5 gift voucher or donate £5 towards research of the same topic on return of both the questionnaire and sample.

Invitation letters were sent in five different batches from each GP practice, with mail-outs at least 1 month apart to facilitate project administration. After each mail-out, stool returns were monitored and the number of invitation letters sent out in later mail-outs, adjusted in the light of the return rate from earlier mail-outs. At some practices all patients within some strata were invited.

## Stool sample kits

If patients were willing to participate, they were asked to return a reply slip with their contact details in a prepaid envelope. Those who returned a positive response reply slip to the invitation were then sent a study information sheet, stool collection kit and questionnaire. The stool sample collection kit had been designed with input from

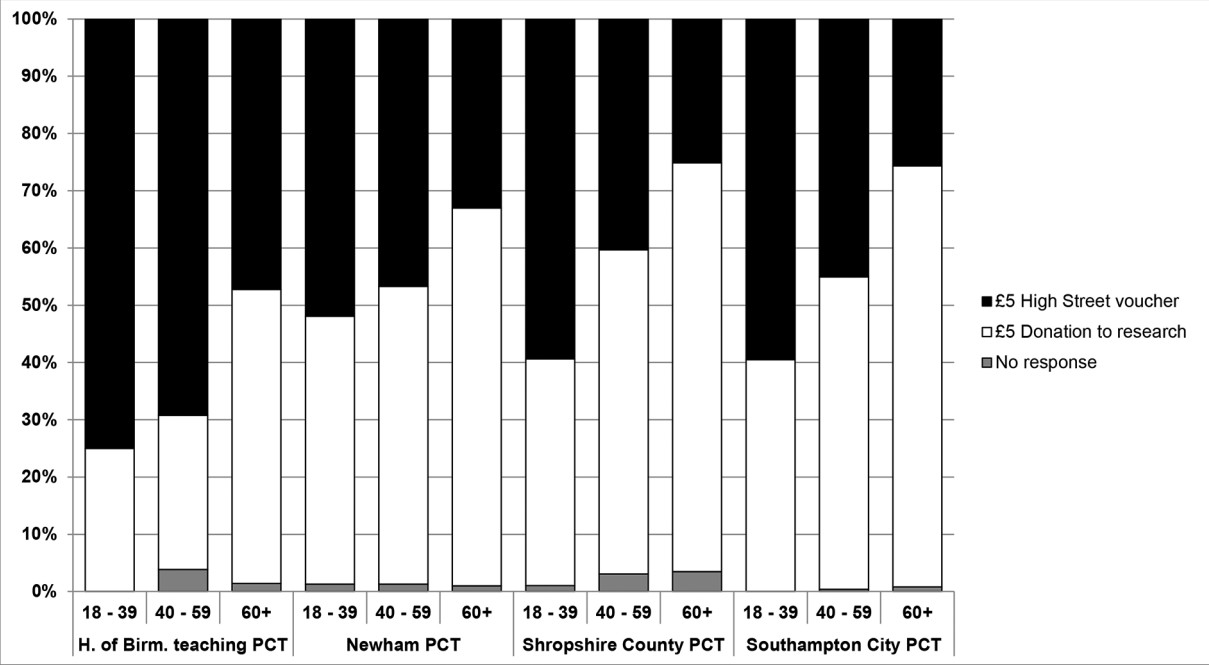

**Figure 3** Incentive options for participants who returned a stool sample and completed questionnaire by age and PCT. PCT, primary care trust.

the general public.[9] Returning the questionnaire and stool sample was taken as implied consent for participation. In addition, willing patients were asked to give written consent to allow the study team to check their medical records for any details on the questionnaire which needed clarifying, and to save their stool sample for future research. The information sheet reiterated information in the invitation letter, and that the results would help the NHS improve the treatment and control of infections in their community and hospital. The stool collection kit was prelabelled with their unique study ID and date of birth and contained a pair of plastic gloves, a sterile 30 mL plastic stool collection pot, pictorial instructions,[9] a spill proof stool pot transporter and a prepaid biological specimen return envelope. Participants were not asked to make any dietary restrictions prior to taking a stool sample; neither were they asked to stop any ongoing medication. Involvement in this study did not entail any visits to the practice or face-to-face contact with the researchers.

Participants were asked to return, by post, the questionnaire, consent form and self-collected stool sample to the research laboratory in the prepaid addressed envelope which fitted into a normal post box. Study flyers at practice receptions, local newspapers and local radio were used to publicise the study. If willing participants did not return the stool sample kit, but provided their telephone number via the invite return slip, researchers made a reminder phone call to ask them to return their samples and questionnaires. To maximise returns, these phone calls were made at different times of day. If necessary a further kit was provided.

### Sample size

Previous research has showed that $bla_{CTX-M}$ extended-spectrum beta-lactamase producing Enterobacteriacae colonisation in diagnostic samples in Birmingham varied from 8.1% in Europeans to 22.8% in Middle East/South Asians.[10] Thus, to have an 80% chance of finding a difference in faecal colonisation between different ethnic groups to be significant the 5% level, assuming the 'true' colonisation percentages were 6% for Europeans and 12% for Asians, a total of 390 in each ethnic group across all four regions giving a total of 1560 participants overall was required. We assumed a 7% overall return rate, and therefore initially planned to send out 20 400 invites.

### Data analysis

Of the 58 337 patients sent a postal invitation, the percentage that returned both a stool sample and a completed questionnaire was calculated—forthwith called stool return rate. We investigated how the stool return rate varied by ethnicity, age group, gender and PCT.

Of the patients sent a postal invitation and returning both a stool sample and a completed questionnaire, we calculated the percentage choosing the £5 gift voucher rather than choosing £5 to be donated towards research on the same topic. If participants ticked both boxes for a £5 gift voucher and for £5 to be donated to research, it was assumed that the participant preferred a voucher. If participants ticked neither box, they were excluded from the analysis of what choice they made.

Participants ticking neither box (for giving consent or not giving consent) for allowing us to access their GP

notes or use their data for future research were assumed to have not given their consent for these two actions.

## RESULTS
### Recruitment

Sixteen practices were recruited to the study: three from Shropshire, four from Newham, five from Southampton and four from Birmingham. Stratifying by ethnicity proved difficult; over 350 ethnic variables were recorded, many of which were ambiguous as descriptions were commonly geographical areas, religions, language spoken and nationality. For the purposes of creating strata based on the GP record of ethnic group, this study created five groups: Asian, black, other/mixed, unknown and white. In total, we recruited 346 Asian patients, 186 black patients, 1709 white patients and 53 mixed/other patients.

### Sample returns

We invited 58 337 patients to participate and 4186 (7.2%) expressed interest. Stool collection kits were sent to 3389 (5.8%) as we stopped sending kits to respondents within the age group over 40 years, and of white ethnicity as when we sufficient numbers of stool samples in these groups; 2388 (70.4%) returned a questionnaire and 2430 (71.7%) returned a stool samples. Overall 2296 (3.9%) returned a complete sample, that is, both a stool sample and completed questionnaire. This included 253 of 535 participants who gave their phone number and were reminded by phone (figure 1). However, we did not reach our goal of obtain 390 samples from each of the four defined ethnic groups.

Returns by PCT, age group, ethnic group and gender were as follows.

### Primary care trust

Complete sample return from invites was 8.6% (762/8885) in Shropshire PCT, 1.6% (152/9385) in Birmingham, 2.9% (583/20 087) in Newham and 3.9% (799/19 980) in Southampton.

### Age group

Complete sample return from invites was 9.9% (994/9960) from patients aged over 60 years, 4.7% (750/15 907) from patients aged 40–59 years and 1.7% (552/32 470) from patients aged under 40 years.

### Ethnic group

Complete sample return from invites was 6.8% (1101/16 181) from white patients, 1.6% (296/18 502) from Asians, 4.1% (171/4146) from blacks, 3.7% (79/2133) from other/mixed and 3.8% (650/17 225) from those of unknown ethnicity. All patients in Shropshire were assumed as being from a white ethnic group.

### Gender

Complete sample returns from female invites was 4.8% (1309/27 540) and 3.2% (987/30 797) from males.

Return rates by PCT, age group and ethnic group are illustrated in figure 2. The highest return rate of 27.3% was in Shropshire (predominately white) in the age group of over 60 years; the lowest, 0.6%, was in Birmingham in the age group of 18–39 years.

*Incentives*: Of participants who returned a completed questionnaire together with a stool sample, 41.6% (942/2261) requested a £5 gift voucher, 57.4% (1319) opted to donate the £5 to research and 1.6% (35) did not indicate a preference. Overall, participants aged 18–39 years (58.5%) preferred a high street voucher whereas those aged over 60 years (69.3%) preferred a donation to research; this was evident across all PCTs (figure 3). Among Indian, Pakistani or Bangladeshi participants 60.0% preferred a £5 gift voucher while among white participants 38.0% preferred a £5 gift voucher. 61.0% of participants in Birmingham requested a voucher, 43.9% in Newham, 38.7% in Shropshire and 39.1% in Southampton.

*Permissions*: 94.9% (2178/2296) of participants who provided a stool and completed questionnaire gave consent for researchers to access their GP notes to clarify any details from the questionnaire. 94.9% (2180/2296) gave permission for their sample and data to be used in future research.

## DISCUSSION
### Return rate

Participation rates in epidemiological studies especially population-based studies have been declining over the years.[11] A US study found that the general public are divided on their willingness to participate in medical research trials; 46% surveyed via telephone said that they would participate in a study for a new treatment for a disease that concerns them, 25% were unwilling and 29% were undecided.[12]

The overall stool and questionnaire return rate from this study (3.9%) were lower than expected, resulting in difficulty in achieving our initial recruitment aim of 390 in each ethnic group; we had planned for a 7.6% return rate. The nature of the sample collection that is, faeces, may have contributed to this low return rate. Previous research examining why patients fail to return stool samples to their GP suggested that 'the taboo associated with the "dirtiness" of human faeces may be a key reason why some people lack the motivation to comply'.[9] However, other research of a similar nature, a gut microbiome study, had a much higher return rate of 20%;[3] however, participants were all aged 55–69 years, were all female, received up to three follow-up phone calls and had stool samples picked up by courier. We also found the return rate for females aged 60 years or more was high but noted that this varied by ethnic group and that reminder phone calls proved particularly beneficial in increasing sample returns; 47% of those contacted returned the specimen.

Our highest return rate in the patients of age over 60 years (20%) is lower than for the BCSP pilot study in England and Scotland where the uptake was 57%–61.8% in patients of age 50–69 years.[13] [14] A point of note is that our research had no personal benefit to the participant while the BCSP pilot study provided further cancer screening and treatment for those screened positive for bowel cancer.

## Incentives

It has been previously noted that participation in research requires motivated individuals[3]; however, the actual motivating factor varies. Offering study results as an incentive does not appear to increase recruitment.[15] Our study offered a £5 gift voucher or donating £5 to research as a potential motivation; of those who opted for a financial incentive, individuals aged 18–39 years are more likely to want a voucher than those aged over 60 years. While we cannot say that a financial incentive was the main motivating factor for young participants, our findings do coincide with other research,[7] [16] suggesting that it may help facilitate recruitment in the younger age groups. However, when factoring this into a research plan, consideration should be given to the fact that the higher the financial incentive the more likely people are to agree to participate.[17]

## Ethnicity

Lower uptake in BME groups compared with whites has been reported elsewhere.[18] The global nature of transmission of multidrug-resistant bacteria[19] emphasises the importance of ethnic minority participation in community surveillance of antimicrobial resistance. Some studies have found that BME groups are more willing to participate if they were approached directly and the research has direct relevance to them.[20] Language and cultural differences have been identified as barriers to recruitment of ethnic minority groups.[4] In each practice, our information sheet has a sentence in the most common non-English languages stating that the information could be provided in those languages; very few foreign language sheets were requested.

## Future consent

Our low return rate suggests that those individuals who did participate may be more motivated than in the normal population, so it is unsurprising that 94.9% of our participants consented to allow researchers to access their GP notes and bank their sample and data for future research. Informing research for future generations has been cited as a motivating factor for consenting to bank samples.[21] Banking samples have been more commonly reported in genetic studies with blood or saliva samples where an over 90% consent rate has also been reported.[22]

## STRENGTHS AND LIMITATIONS

This was a large multicentre community-based study that included adult participants of variable age groups, gender and ethnicities from four areas in England. The majority of the Asians in our study were from Birmingham and mostly spoke Urdu. While we cannot categorically say that Asians from other areas of the Indian subcontinent would have similar low returns, other research involving stool returns has described uptake as *strikingly low* in ethnically diverse populations.[2] As we have invited a large number of patients from different ethnicities to participate, we feel that our return rates are likely to be comparable in future studies requesting stool samples from the general population.

Recruiting patients in batches at each practice was a strength of the sampling design because it allowed us to compensate for the lower than expected return rate by increasing invited in cohorts with lower returns.

It could be argued that there are two different results that should be reported—willingness to participate is very low—7%, but the return rate of samples is higher, 67.7% because 2296/3389 patients who were sent the collection kit returned complete sample and not the 3.9% (2296/58 337) we report. However, we feel that by only reporting the return rate from those who expressed interest could be viewed as biased as we would be looking at those participants who are obviously interested in participating and would vastly underplay the amount of effort it requires to obtain a sufficient sample size from the general population.

Use of a stool collection instruction leaflet and a prepackaged stool kit delivered to participant's homes may have aided compliance and stool returns.[23] In addition, returning stool specimens by post had the advantage of reducing perceived embarrassment of returning a stool sample to a GP Practice receptionist.[9] Ethics permitted only anonymous patient information be removed from practices meaning researchers could not follow-up with participants who did not respond to the initial invite; this is unfortunate as follow-up phone calls and interaction with the research team encourage higher recruitment rates.[7] [24] Patients received letters from their GP practice but were asked to send samples to a different location; this may have been confusing to some patients.

## CONCLUSIONS

The low stool specimen return rate and its wide variation by ethnicity and age has implications for future studies that involve the collection of stool specimens from the general population and have no health benefit to their participants. Unless measures are taken to counteract this variation in the return rate, samples will tend to under-represent Asians and younger individuals. Furthermore, research is needed to explore how to maximise stool return rates in research. Other forms of recruitment (other than postal recruitment) might be effective at increasing the return rate; however, if postal is the recruitment method of choice, then reminder phone calls are recommended. Increasing the value of the gift voucher could be effective at increasing the return rate but obviously this increases the cost of the study and also risks introducing a new selection bias.

**Author affiliations**
¹Public Health England, Primary Care Unit, Gloucester, UK
²Statistics, Modelling and Bioinformatics Department, Public Health England, London, UK
³Public Health Laboratory, Heart of England NHS Foundation Trust, Birmingham, UK
⁴University of Southampton Faculty of Health Sciences, Southampton, UK
⁵TB Surveillance Unit, Public Health England, London, UK
⁶Midlands & NW Bowel Cancer Screening Hub, Coventry, UK

**Acknowledgements** We are grateful for the support received from the Comprehensive Local Research Network, Clinical Research Network: Primary Care, Primary Care Unit Staff, Study GPs and Practice managers in the Birmingham area, Southampton, Shropshire and Newham.

**Contributors** DL (March-July 2014 and from April 2015) was involved in data collection and data management, was a steering group member. DN-S was involved in ethics application, practice and participant recruitment, data collection and entry. DL and DN-S contributed equally to this paper . TN was grant co-applicant, involved in study design, practice and participant selection, data management, data analysis, was a steering group member, and contributed to the writing of the manuscript. PH was grant co-applicant and involved in literature review, study design, questionnaire design, laboratory supervision, data interpretation, was a steering group member and contributed to the writing of the manuscript. KT involved in participant recruitment and liaison, data collection and entry, and agreed the final manuscript. MT was grant co-applicant and involved in study and questionnaire design, Steering Group member, Primary Care Lead, practice selection and commented on the manuscript. HLT was grant co-applicant and involved in study and questionnaire design, data interpretation, was a steering group member and commented on the manuscript. LX-M was a grant co-applicant and involved in study design, laboratory supervision, supported laboratory data management, was a steering group member and contributed to the writing of the manuscript. SSh involved in laboratory work and data collection, and agreed the final manuscript. SM involved in laboratory work and data collection, and agreed the final manuscript. AA-B was involved in laboratory work and data cleaning, and agreed the final manuscript. SSm was grant co-applicant and involved in study design, was a steering group member, and commented on the manuscript. K-TC involved in laboratory work, recording and data entry and agreed the final manuscript. CM led the writing of the grant application and protocol, was involved in the literature review, contributed to the design of the questionnaire, led the project steering group and led the writing of the manuscript.

**Funding** The report is based on independent research commissioned andfunded by the NIHR Policy Research Programme (Ref. 041/0038S). The viewsexpressed in the publication are those of the author(s) and not necessarilythose of the NHS, the NIHR, the Department of Health, 'arms' length bodies orother government departments.

**Competing interests** All findings and observations are original unless otherwise acknowledged.

**Patient consent** Detail has been removed from this case description/these case descriptions to ensure anonymity. The editors and reviewers have seen the detailed information available and are satisfied that the information backs up the case the authors are making.

**Ethics approval** Ethical approval for the study was obtained from the NRES Committee South West -Frenchay, Bristol, UK (13/SW/0017). The data we collected from GP practices were anonymous.

**Provenance and peer review** Not commissioned; externally peer reviewed.

**Data sharing statement** The relevant anonymised patient level data are available on reasonable request from the authors.

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
