## [Reviewer comments · BMJ Open]

ARTICLE DETAILS

TITLE (PROVISIONAL)	Informing future research for carriage of multi-resistant gram negative bacteria: Problems with recruiting to an English stool sample community prevalence study.
AUTHORS	Lecky, Donna; Nakiboneka-Ssenabulya, Deborah; Nichols, Tom; Hawkey, Peter; Turner, Kim; Chung, Keun-Taik; Thomas, Mike; Thomas, Helen; Xu McCrae, Li; Shabir, Sahida; Manzoor, Susan; Alvarez-Buylla, Adela; Smith, Steve; McNulty, Clodna

VERSION 1 – REVIEW

REVIEWER	Miroslav Fajfr Institute of Clinical Microbiology University Hospital in Hradec Kralove Sokolska 581 Hradec Kralove 50005 Czech Republic
REVIEW RETURNED	11-Aug-2017

GENERAL COMMENTS	The study is very interesting and very needing, as the findings of this study need to be taken into account when planning future studies on stool specimen collection (e.g. study of gut microbiota). The manuscript is clearly structured and the design of the study is well described, including a comprehensive description of patient selection and number of patient included in study. The authors clearly state the strengths and weaknesses of the study. The results of the study are clearly presented and discussed in the relevant parts of the manuscript. The used statistical methods are clearly described. Literary sources are adequate and up-to-date on the topic. However, I have a few comments on the article: 1. I personally believe that the low return rate is related to the material itself - faeces, because some individuals (ethnicity) may have an internal problem with handling this type of sample (collecting and transporting). For other samples, I expect a higher return rate.2. The plan was to obtain 390 samples from each of the 4 defined ethnic groups (page 9, line 14), but this goal was not fulfilled. Only 346 samples from the Asian population, 186 from Black patients, 53 from mixed / other patients but 1709 White patients' samples were obtained (page 10, line 24-26). I suggest to make the short mention in the text that this goal has not been fulfilled. Possible reasons for this are then well discussed on page 13.
--

	3. I see the main lack of the article in the uncertain return rate calculation for me. If I correctly understand the design of the study (page 8): a large group of patients was invited to participate into study, with a expected positive response to 7%. Only these patients are then sent with information, questionnaire and stool sample collection kits. The Sample return (page 10) therefore describes: "We have invited 58,337 patients to participate and 4,186 (7.2%) have expressed interest. Stool collection kits were sent to 3,389 (5.8%), ..." My questions: Why the stool collection kits were sent to only 3,389 patients and not to all people who want to participate? Why the percentage of the returned sample / questionnaire was not calculated by people who expressed interest, respectively from whom were sent the collection kits? According to my opinion it is not correct to make the calculation from the number of invited patients. The patients who do not need to participate can not send the samples / questionnaire, logically. There are two different results - willingness to participate is very low - 7%, but the return rate of samples is higher, because 2,296 patients from 3,389 (whom were sent the collection kit) returned complete sample. So, according to my understanding, the return rate is 67.7% (2,296/3,389) and no 3.9% (2,296/58,337). 4. Please use unify all percentages with one decimal place
--	---

VERSION 1 – AUTHOR RESPONSE

The study is very interesting and very needing, as the findings of this study need to be taken into account when planning future studies on stool specimen collection (e.g. study of gut microbiota). The manuscript is clearly structured and the design of the study is well described, including a comprehensive description of patient selection and number of patient included in study. The authors clearly state the strengths and weaknesses of the study. The results of the study are clearly presented and discussed in the relevant parts of the manuscript. The used statistical methods are clearly described. Literary sources are adequate and up-to-date on the topic. The authors would like to thank the reviewer for their time and constructive comments which we have addressed as outlined below.

However, I have a few comments on the article:

1. I personally believe that the low return rate is related to the material itself - faeces, because some individuals (ethnicity) may have an internal problem with handling this type of sample (collecting and transporting). For other samples, I expect a higher return rate.

Response: We agree that this may have been a contributing factor and have therefore added the following sentence to the "Discussion and return rate section" The nature of the sample collection i.e. faeces, may have contributed to this low return rate. Previous research examining why patients fail to return stool samples to their GP suggested that the taboo associated with the 'dirtiness' of human faeces may be a key reason why some people lack the motivation to comply.⁹

2. The plan was to obtain 390 samples from each of the 4 defined ethnic groups (page 9, line 14), but this goal was not fulfilled. Only 346 samples from the Asian population, 186 from Black patients, 53 from mixed / other patients but 1709 White patients' samples were obtained (page 10, line 24-26). I suggest to make the short mention in the text that this goal has not been fulfilled. Possible reasons for this are then well discussed on page 13.

Response: We agree and have added the following sentence in the Results and sample returns section However, we did not reach our goal of obtain 390 samples from each of the 4 defined ethnic groups.

3. I see the main lack of the article in the uncertain return rate calculation for me. If I correctly understand the design of the study (page 8): a large group of patients was invited to participate into study, with a expected positive response to 7%. Only these patients are then sent with information, questionnaire and stool sample collection kits. The Sample return (page 10) therefore describes: "We have invited 58,337 patients to participate and 4,186 (7.2%) have expressed interest. Stool collection kits were sent to 3,389 (5.8%), ..."

My questions:

a. Why the stool collection kits were sent to only 3,389 patients and not to all people who want to participate?

In the methods section we state that Towards the end of the stool collection period we stopped sending kits to respondents within the over 40 year age group, and of white ethnicity as we had reached sufficient numbers of stool samples in these groups however to avoid confusion, I have moved this to the results section.

b. Why the percentage of the returned sample / questionnaire was not calculated by people who expressed interest, respectively from whom were sent the collection kits?

According to my opinion it is not correct to make the calculation from the number of invited patients. The patients who do not need to participate cannot send the samples / questionnaire, logically. There are two different results - willingness to participate is very low - 7%, but the return rate of samples is higher, because 2,296 patients from 3,389 (whom were sent the collection kit) returned complete sample. So, according to my understanding, the return rate is 67.7% (2,296/3,389) and no 3.9% (2,296)

Response: The reviewer makes a very good point regarding willingness to participant and return rate; however we feel that the emphasis needs to be placed on the number of participants one needs to approach in order to obtain an expression of interest. The aim of this paper is to inform future research of the difficulties in recruiting to a study requiring stool samples; only reporting the return rate from those who expressed interest could be viewed as biased as we would be looking at those participants who are obviously interested in participating and would vastly underplay the amount of effort it requires to obtain a sufficient sample size from the general population. We have added words to this effect in the strengths/limitations section of the paper.

4. Please use unify all percentages with one decimal place

Response: This has been done throughout the text when reporting our findings however we could not change when citing other work and these are the figure provided in the cited papers.

VERSION 2 – REVIEW

REVIEWER	Miroslav Fajfr, MD, PhD Institute of Clinical Microbiology University Hospital Hradec Kralove Czech Republic
REVIEW RETURNED	09-Nov-2017
GENERAL COMMENTS	The study is very interesting and very needing, as the findings of this study need to be taken into account when planning future studies on stool specimen collection (e.g. study of gut microbiota). The manuscript is clearly structured and the design of the study is well described, including a comprehensive description of patient selection and number of patient included in study. The authors clearly state the strengths and weaknesses of the study. All my previous questions and commentaries were accepted or discussed. I recommend this work for publication.